# Anaerobic Digestion of Synthetic Municipal Wastewater (MWW) in a Periodic Anaerobic Baffled Reactor (PABR): Assessment of COD Removal and Biogas Production

Achilleas Zarkaliou [1], Christos Kougias [1], Anna Mokou [1], Konstantina Papadopoulou [1,*] and Gerasimos Lyberatos [1,2]

[1] School of Chemical Engineering, National Technical University of Athens, Iroon Polytechneiou 9, Zografou, 15780 Athens, Greece

[2] Institute of Chemical Engineering Sciences (ICE-HT), Stadiou Str., Platani, 26504 Patras, Greece

* Correspondence: kpapado@chemeng.ntua.gr

**Abstract:** The benchmark approach for municipal wastewater treatment is based on biological oxidation. Due to high energy consumption, alternative treatment schemes are proposed, among which anaerobic digestion is the most promising. In this work, the direct anaerobic digestion of municipal wastewater in a high-rate system is examined. The reactor utilized for the study is the periodic anaerobic baffled reactor (PABR). Two distinct experimental cycles were conducted, during which the operational parameters of the PABR were consecutively modified: in the first cycle, six phases were conducted where the hydraulic retention time (HRT) varied from 10 to 1 days, the period T between 2.5 days and 0.25, while the OLR remained constant at values near 1.0 gsCOD/L/d. During the second cycle, four distinct phases were conducted with no switching imposed. The HRT varied from 4 to 1 d. The last experimental phase of both cycles was the most significant, due to feedstock resemblance to raw wastewater. The biogas and the biomethane production rates reached 66.8 L/d and 41.1 L/d, respectively, while the COD reduction rate reached 73.7%. Conclusively, the PABR is a high-rate AD system, capable of treating MWW under extreme operational conditions.

**Keywords:** high-rate anaerobic digestion; municipal wastewater; PABR

## 1. Introduction

Wastewater treatment is a key process in ensuring public health and environmental well-being. The environmental aspect of this process is based on removing pollutants, such as organic matter and nutrients (N, P), before disposal. The energy cost of the activated sludge process, the most widely used municipal wastewater treatment process, is quite high (0.69–3.01 kWh/kg COD) [1]. The increase in the cost of energy has led to research on more environmental and economically efficient methods. The new approaches of domestic wastewater management are based on circular ways to recover valuable nutrients and energy, as opposed to the traditional linear methods [2]. Currently, the benchmark approach to municipal wastewater management consists of sewer collection, treatment in a facility aimed at removal of suspended solids through primary sedimentation, biological oxidation of organic matter under aerobic conditions, biological nutrient (N and P) removal, and disposal of the clarified effluent following disinfection by chlorination. The process generates a mixture of primary and excess secondary sludge, which are typically mixed, stabilized by anaerobic digestion, and dewatered before disposal [3]. The key operating costs lie in the aeration and in sludge (biosolids) management [4]. Anaerobic digestion emerges as a feasible solution that aims to reduce the energy requirements of WW treatment and is already globally implemented in WW treatment plants. However, AD is most commonly used as a side process in WWTPs, mostly for stabilization of the sludge generated during primary and secondary sedimentation. The efficiency of direct AD of

MWW has been proposed by many researchers, and notable is the installment of a full-scale expanded granular sludge bed reactor (EGSBR) in a WWTP in Ireland, during the operation of which a high BOD removal rate (85%) was accomplished [5]. Another management scheme that is a substantial departure from the conventional activated sludge process is the low-energy mainline (LEM) process [6]. This scheme is based on direct low-strength anaerobic digestion of municipal wastewater after primary sedimentation and use of the reactor's effluent as irrigation water, as no significant amount of nitrogen and phosphorus would be consumed in the anaerobic process, based on the same concept by examining the capabilities of an innovative high-rate anaerobic system to treat municipal wastewater.

The aim of this work is to examine the suitability of a novel high-rate anaerobic system—the periodic anaerobic baffled reactor (PABR)—to treat municipal wastewater directly.

## 2. Materials and Methods

### 2.1. Experimental Process

The PABR is a high-rate anaerobic system designed by Skiadas and Lyberatos [7]. It consists of two concentric cylinders, as shown in Figure 1, with the inner one operating as heat exchanger. The space between the two cylinders is divided into four compartments, each of which is divided in two sections, one downflow and one upflow, thus resembling a simple anaerobic baffled reactor (ABR), only arranged in a circular structure.

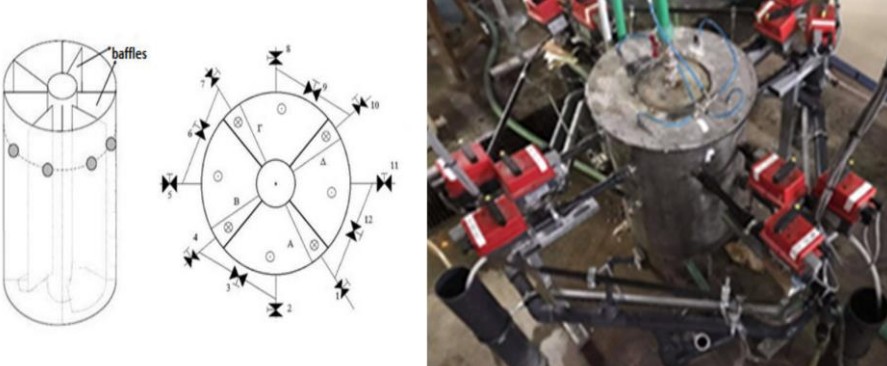

**Figure 1.** Experimental setup of the PABR used in the experiments. A,B,Γ,Δ indicate the compartments of the reactor. Valves 1,4,7,10 are the inlet valves of compartments A,B,Γ,Δ while valves 2,5,8,11, are their effluent valves. Valves 3,6,9,12 control the communication between the compartments.

The innovative approach of this bioreactor is its ability to periodically change the inflow and outflow compartments. The PABR introduces a novel operational parameter called switching period (T), which along with the hydraulic retention time (HRT) rearranges the flow patterns of the reactor. T is the period for one complete switching of compartment roles. Through this parameter, the time each compartment will be used as the inlet and the outlet is determined. The switching between the compartments is achieved with valves fitted on the external piping, which determine whether the compartment will be used as the inlet or outlet or as an intermediate compartment. This flexibility gives the biomass the opportunity to withstand fluctuations of the feed concentration, leading to easier culture adaptation under extreme or varying conditions.

When the HRT/T ratio is equal to 1, every compartment receives the same amount of untreated substrate. On the other hand, if we do not switch periodically the role of each compartment, the flow pattern of the reactor resembles that of a four-compartment anaerobic baffled reactor. A pilot-scale 77L active volume PABR was utilized and both operation modes (switching and ABR mode) were examined and evaluated for similar organic loading rates and HRTs.

A number of studies have examined the performance of the PABR under various OLRs, HRTs, and TS and by utilizing different types of feedstock other than typical municipal wastewaters [8–12].

### 2.2. Feedstock Composition

In order to secure a constant composition of the feed, instead of real raw municipal wastewater, a synthetic mixture was used that resembles its main characteristics. The synthetic wastewater used for the PABR in the present study consists of: 10 to 1.0 g/L glucose, 0.306 to 0.0285 g/L $NH_4Cl$ (regarding the experimental phase), 0.08 g/L $CH_3COONa$, 0.044 g/L $KH_2PO_4$, 0.0275 g/L $MgSO_4 \cdot 7H_2O$, 0.0025 g/L $CaCl_2$, 0.004 g/L KCl, 0.125 g/L $NaHCO_3$, 1.875 mg/L $FeCL_3 \cdot 6H_2O$, 0.1875 mg/L $H_3BO_3$, 0.225 mg/L KI, 0.15 mg/L $MnSO_4$, 0.0275 mg/L $ZnSO_4 \cdot 7H_2O$, 0.0375 mg/L $CuSO_4 \cdot 5H_2O$ and 12.5 mg/L EDTA [13]. Every experimental phase had different concentration of glucose and ammonium chloride so that the OLR would be kept constant while reducing the HRT. The relative concentrations of these two substances were chosen so that the C/N ratio was maintained constant close to 50 [14].

### 2.3. Analytical Methods

The scope of the experimental process was to evaluate the efficiency of the PABR under different conditions in terms of organic load reduction and biogas and biomethane productivity. Initially, the COD was high (10,000 mg/L) in order to achieve good anaerobic digestion operation and then, as the municipal wastewater typically has COD concentrations under 1000 mg/L, the organic load of the feed mixture was progressively reduced while keeping the organic loading rate constant. Therefore, the bioreactor operated under various HRTs and T while the organic loading rate was kept at values near 1 gsCOD/L/d (as outlined in Tables 1 and 2).

**Table 1.** Operational parameters imposed during the first experimental cycle.

| First Experimental Cycle | Experimental Phases | | | | | |
|---|---|---|---|---|---|---|
| Operational Parameters | First | Second | 3rd | 4th | 5th | 6th |
| Duration (d) | 21 | 33 | 20 | 10 | 69 | 9 |
| HRT (d) | 10 | 6 | 4 | 3 | 2 | 1 |
| T (d) | 10 | 6 | 4 | 3 | 2 | 1 |
| OLR (g sCOD/L/d) | 0.91 | 0.96 | 0.95 | 0.90 | 0.91 | 1.06 |

**Table 2.** Operational parameters imposed during the second experimental cycle.

| Second Experimental Cycle | Experimental Phases | | | |
|---|---|---|---|---|
| Operational Parameters | First | Second | 3rd | 4th |
| Duration (d) | 28 | 15 | 15 | 22 |
| HRT (d) | 4 | 3 | 2 | 1 |
| OLR (gCOD/L/d) | 0.97 | 0.97 | 1.02 | 1.06 |

The reactor was fed with the synthetic wastewater and operated under mesophilic conditions (35 °C), for 242 consecutive days. Two distinct experimental cycles were conducted, as shown in Tables 1 and 2. During the first cycle, six consecutive experimental phases were conducted, with the reactor operating in a switching mode with constant HRT/T and OLR/HRT ratios. During the second experimental cycle, four experimental phases were conducted with similar OLR/HRT ratios, but the reactor operated in the ABR mode with no switching imposed. Throughout the experimental process, pH, total alkalinity, total suspended solids (TSS), volatile suspended solids (VSS), soluble chemical oxygen demand (sCOD), biogas production and methane content were monitored at regular intervals. TSS, VSS, sCOD and alkalinity were measured according to standard methods (APHA, 1995),

while a GC-TCD (Shimadzu GC-2014, Duisburg, Germany) was used for the measurement of the methane content of the generated biogas.

## 3. Results and Discussion

### 3.1. First Experimental Cycle

The overall efficiency of the PABR throughout the whole experimental cycle is presented in Table 3. The HRT reduction from 10 d to 1 d affected the biogas yield, as well as the COD removal achieved. Furthermore, the experimental phase with the lowest HRT showed the highest amount of biogas (1d HRT—44.3 L/d) and biomethane production (1d HRT—26.5 L/d). On the other hand, the average COD removal rate during the HRT 1 d phase is reduced to 69.4%. The highest COD removal occurred during the fourth experimental phase (HRT 3 days) where the COD consumption of the process reached 89%. As shown in Table 3, during all experimental phases the PABR effluent had significant deviations in COD concentrations.

**Table 3.** Average daily results of the first experimental cycle phases.

| First Experimental Cycle | Experimental Phases | | | | | |
|---|---|---|---|---|---|---|
| Characteristics | 1st | 2nd | 3rd | 4th | 5th | 6th |
| Biogas Production (L/d) | 25.6 ± 4.2 | 33.8 ± 3.7 | 37.0 ± 3.2 | 35.6 ± 2.5 | 32.3 ± 4.9 | 44.3 ± 1.8 |
| Biomethane Production (L/d) | 6.9 ± 3.4 | 15.2 ± 5.4 | 18.7 ± 3.7 | 18.5 ± 5.8 | 21.1 ± 4.2 | 26.5 ± 1.6 |
| $CH_4$ Biogas content (%) | 27.0 ± 10.7 | 44.5 ± 13.4 | 50.9 ± 10.5 | 54.3 ± 17.6 | 66.5 ± 8.4 | 59.6 ± 0.6 |
| sCOD Feedstock (g/L) | 9.11 ± 0.75 | 5.74 ± 0.23 | 3.78 ± 0.59 | 2.71 ± 0.29 | 1.81 ± 0.16 | 1.06 ± 0.06 |
| sCOD Effluent (g/L) | 1.86 ± 1.20 | 0.94 ± 0.94 | 0.50 ± 0.91 | 0.28 ± 0.12 | 0.25 ± 0.15 | 0.37 ± 0.12 |
| Average COD removal (%) | 79.5 | 83.7 | 86.9 | 89.6 | 85.30 | 64.9 |

As shown in Figure 2, the daily biogas production increased as the HRT was reduced from 10 days to 6 days and remained at the same daily production levels when the HRT took the values of 4, 3 and 2 d. During the last experimental phase, the average daily biogas increased to 44.3 L/d. The biomethane production is calculated by the concentration of methane in the daily generated biogas. Generally, it is concluded that as the HRT is lowered, the methane concentration of biogas increases, with the exception of the last experimental phase, as shown in Figure 3. In Figure 3, the daily concentration of methane in biogas is presented. It is clear that as the HRT is lowered, the methane content of the biogas increases.

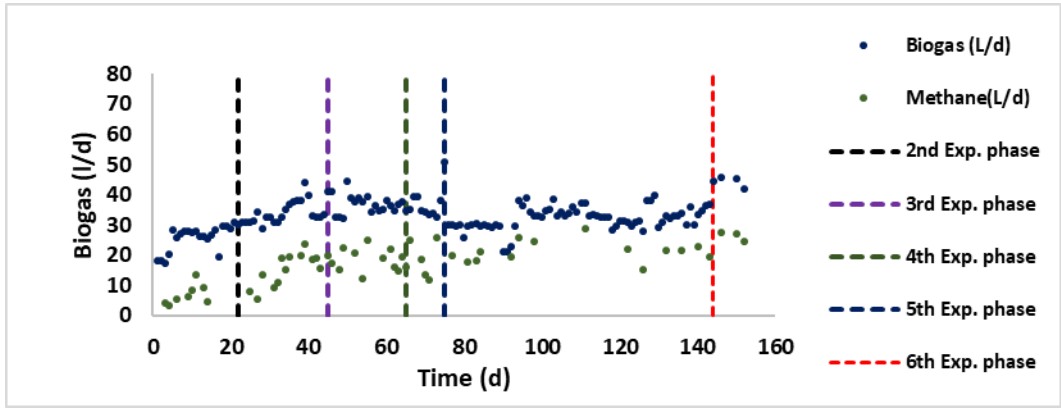

**Figure 2.** Daily biogas and biomethane production of PABR.

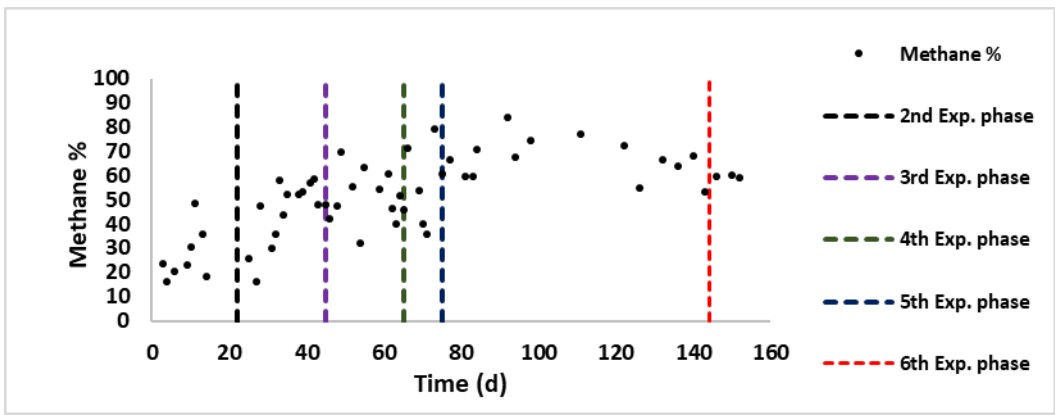

**Figure 3.** Daily methane concentration of the biogas generated in the PABR.

Regarding the behavior of the process in removing the organic content of the waste, two aspects are examined. First, the COD removal is examined between the feedstock and the effluent of the reactor, and second, the concentration distribution of the COD between the four compartments of the PABR. Figure 4 shows the variation of sCOD with time during the consecutive six phases of the PABR switching mode operation. In all cases, the process shows significant sCOD reduction. The only experimental phase presenting significant sCOD at the effluent is the first experimental phase (HRT 10 d). This happens due to the time required for acclimation to the new feedstock type. The inoculum used for the PABR startup was obtained from a CSTR reactor operating in higher HRTs and with excess activated sludge as feedstock. By observing the COD reduction of the last experimental phase (HRT 1 d), where the synthetic waste resembles as much as possible the characteristics of MWW, it is concluded that although the effluent is not yet directly disposable due to its COD concentration, a significant reduction is achieved without consuming the equivalent energy needed for aeration in biological oxidation.

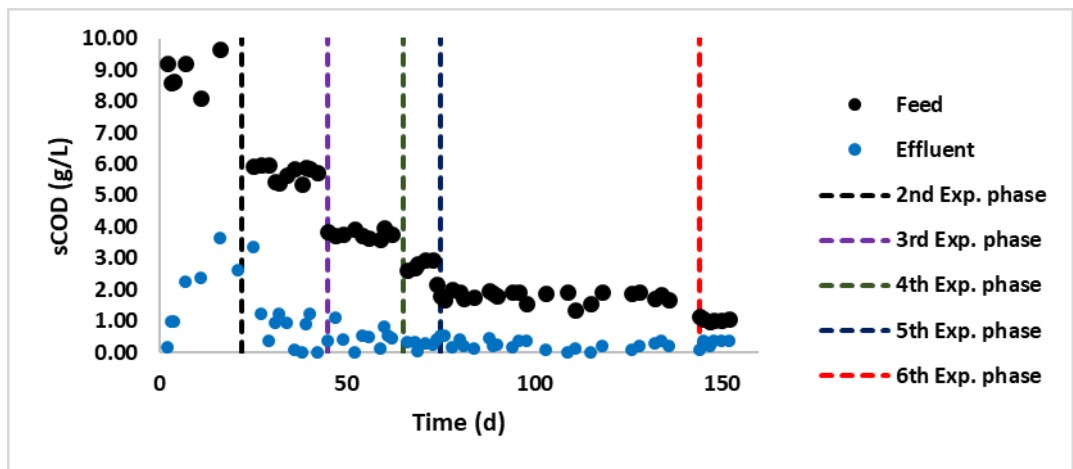

**Figure 4.** Daily sCOD of the reactor's feedstock and effluent.

Figure 5 shows the average sCOD concentration and also the variation in sCOD in the four compartments of the PABR. It has been demonstrated that when switching is frequent enough (low switching period T), the PABR resembles the behavior of a UASBR (upflow anaerobic sludge blanket reactor) [7]. This was indeed verified also in this case when operating with a constant HRT/T ratio.

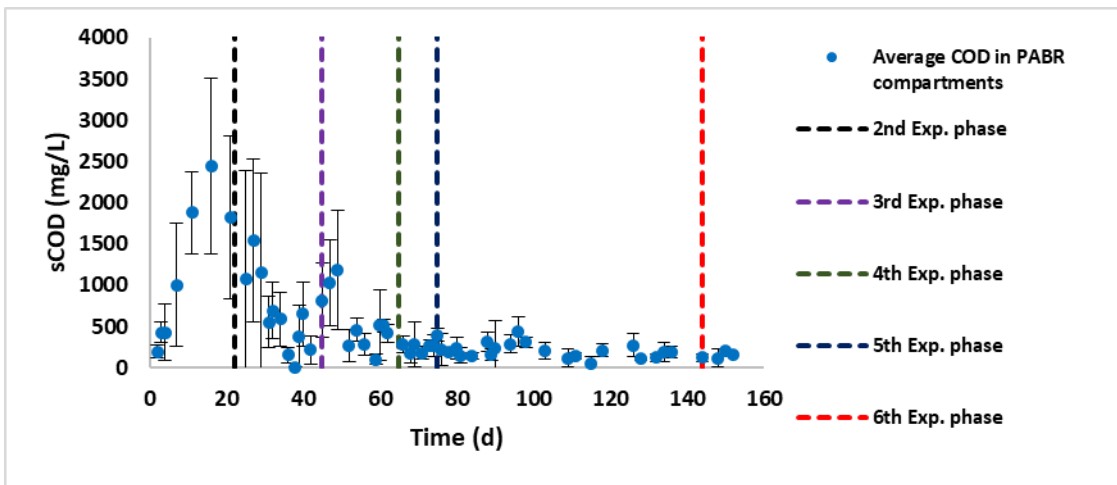

**Figure 5.** Average daily sCOD concentration and variation of in the four PABR compartments.

Similar variation in the COD concentration was also noticed for other process parameters, such as the pH and the total alkalinity, as shown in Figures 6 and 7, respectively. The pH values have strong fluctuations during the higher HRT phases and after the third experimental phase are stabilized at values between 7 and 8. Due to the characteristics of the feedstock, the pH did not cause inhibition of the process as the pH of the feedstock is in the favorable range for anaerobic digestion. On the other hand, the alkalinity of the synthetic MWW proposed by the literature is significantly low (0.125 g/L $NaHCO_3$ = 78mg $CaCO_3$/L), and due to the water used in feedstock preparation, the total alkalinity of the feedstock was 913 mg$CaCO_3$/L in the first experimental phase. As the HRT is lowered, the low alkalinity of the feedstock could introduce a limitation for the process. In order to avoid this problem, the proposed alkalinity of the synthetic mixture was changed and more $NaHCO_3$ was added as alkalinity buffer in order to stabilize the alkalinity of the process above the value of 2000 mg$CaCO_3$/L. As the total alkalinity of the feedstock increased, the alkalinity of the reactor's compartments increased also. During the first experimental phase feeding the reactor with 912.50 mg$CaCO_3$/L on average, the compartments average alkalinity was 1147 mg/L. Increasing the alkalinity of the feedstock to 2500 mg$CaCO_3$/L led to an average alkalinity of 1821 mg $CaCO_3$/L in the reactor. Further increase in the feedstock alkalinity led to higher increase inside the reactor compartments, with the highest amount noticed during the fourth experimental phase (3560 mg $CaCO_3$/L). It is observed that as the alkalinity took higher values, the productivity of the reactor in biogas generation and biomethane concentration increased. This is an indication of the significant role of alkalinity for process efficiency. It can be concluded that if high-rate anaerobic digestion is to replace the activated sludge process for MWW treatment, the alkalinity of the feedstock can be a major limitation parameter for the process. In Table 4, the average daily pH and total alkalinity values are given for the reactor, the feedstock and the effluent.

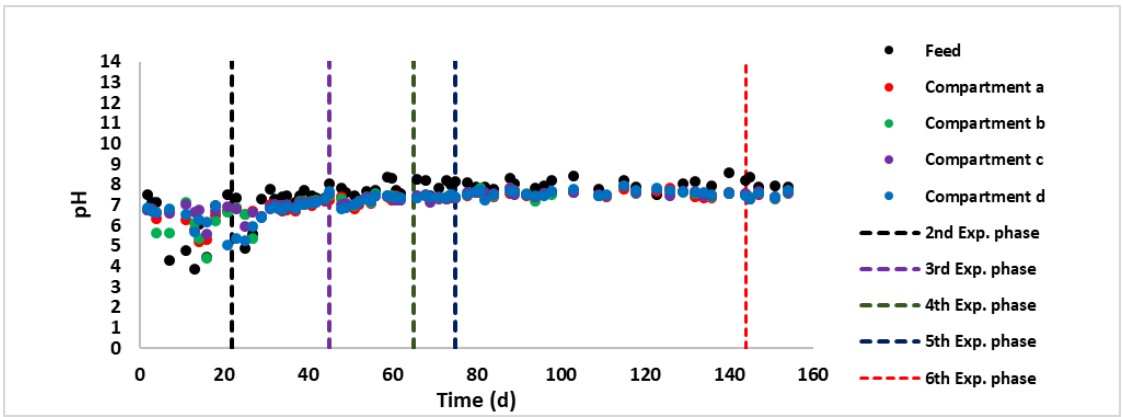

**Figure 6.** Daily pH values of feedstock and reactor compartments.

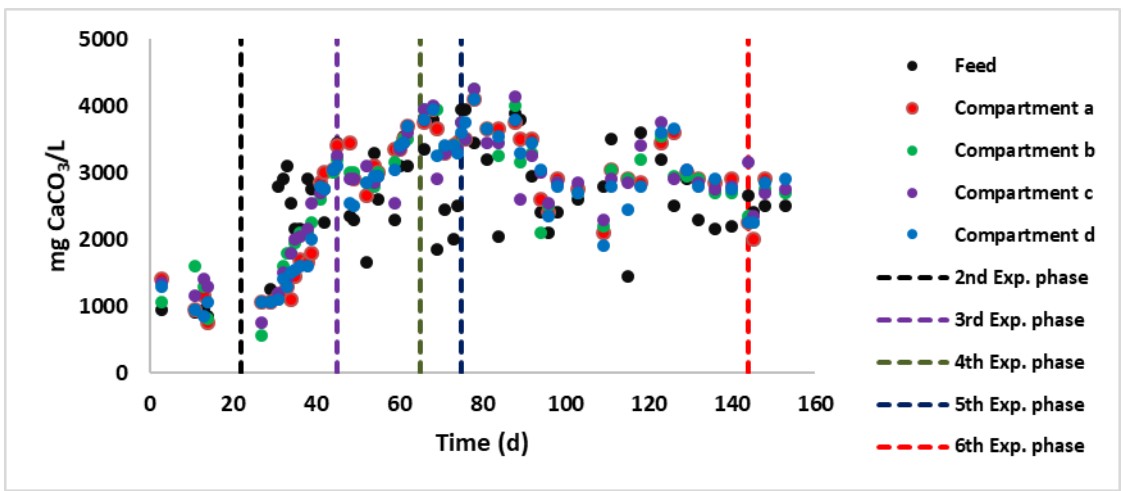

**Figure 7.** Daily total alkalinity value of the feedstock and reactor compartments.

**Table 4.** Average daily pH and total alkalinity of the feedstock, compartments and effluent.

| | Feedstock | | Compartment a | | Compartment b | | Compartment c | | Compartment d | | Effluent | |
|---|---|---|---|---|---|---|---|---|---|---|---|---|
| Exp. Phase | pH | Alkalinity mgCaCO$_3$/L | pH | Alkalinity mgCaCO$_3$/L | pH | Alkalinity mgCaCO$_3$/L | pH | Alkalinity mgCaCO$_3$/L | pH | Alkalinity mgCaCO$_3$/L | pH | Alkalinity mgCaCO$_3$/L |
| 1st | 5.97 | 912.5 | 6.23 | 1063 | 6.00 | 1188 | 6.64 | 1300 | 6.53 | 1038 | 6.26 | 1147 |
| 2nd | 7.11 | 2500 | 6.87 | 1719 | 6.86 | 1896 | 6.89 | 1919 | 6.49 | 1750 | 6.87 | 1821 |
| 3rd | 7.75 | 2770 | 7.23 | 3240 | 7.31 | 3140 | 7.27 | 3090 | 7.26 | 3050 | 7.36 | 3130 |
| 4th | 8.00 | 2658 | 7.36 | 3583 | 7.42 | 3667 | 7.39 | 3475 | 7.38 | 3517 | 7.51 | 3560 |
| 5th | 7.94 | 2880 | 7.55 | 3555 | 7.56 | 3095 | 7.59 | 3145 | 7.60 | 3127 | 7.65 | 3130 |
| 6th | 8.15 | 2512 | 7.46 | 2125 | 7.51 | 2300 | 7.49 | 2738 | 7.44 | 2563 | 7.53 | 2431 |

Regarding the concentration of the biosolids, it seems that the solid content in the PABR compartments was reduced during the experiment. This phenomenon can lead to the assumption that as the HRT is reduced, the reactor reduces its biomass. However, the low concentrations of suspended solids and volatile suspended solids in the compartments, the effluent is caused due to biomass sedimentation in the reactor lower parts, and this is typical for a high-rate system such as this. As shown in Figure 8, in each PABR compartments a two-phase system is formed between the biomass (solid phase) and the treated effluent (liquid phase). The sampling and the effluent valve of each compartment are placed in the higher parts of the reactor.

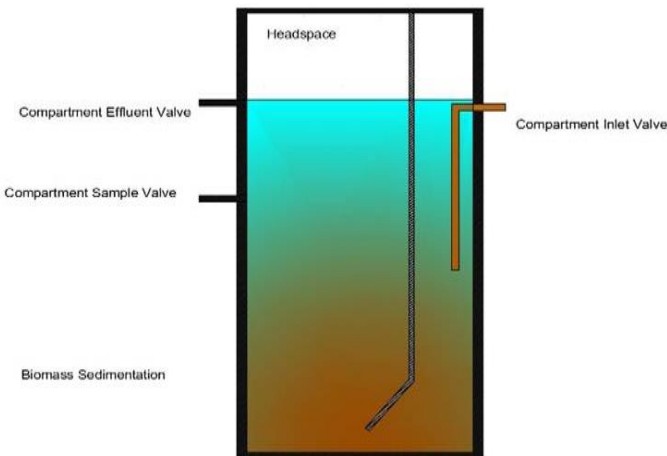

**Figure 8.** PABR compartment cut.

In Figures 9 and 10, the daily concentrations of the TSS and VSS measured from samples obtained from the four separate compartments are presented. In general, we can conclude that as the HRT is reduced, both TSS and VSS are also reduced. In the first experimental phase (10 d HRT), the average TSS and VSS concentrations were 0.83 g/L and 0.63 g/L respectively, while in the last experimental phase the same concentrations were 0.22 g/L and 0.14 g/L, respectively. In Table 5, the average TSS and VSS concentration of every compartment and that of the effluent for all experimental phases is presented.

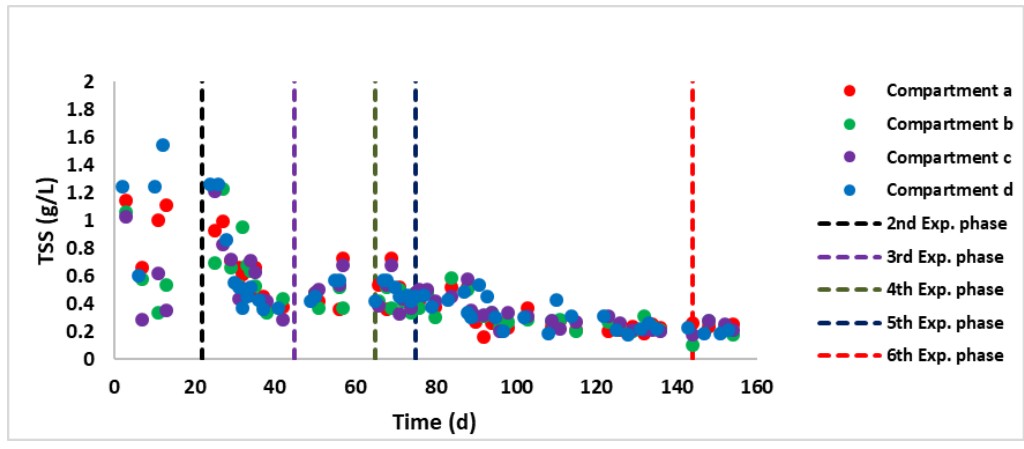

**Figure 9.** Daily TSS concentration in reactor compartments.

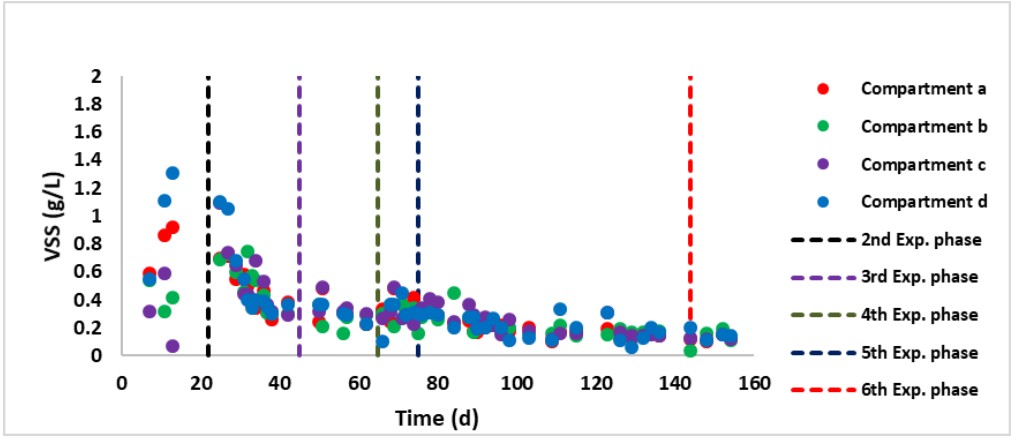

**Figure 10.** Daily VSS concentration in reactor compartments.

**Table 5.** Average daily TSS and VSS concentration in feedstock, reactor's compartments and effluent.

| Exp. Phase | Compartment a TSS (g/L) | Compartment a VSS (g/L) | Compartment b TSS (g/L) | Compartment b VSS (g/L) | Compartment c TSS (g/L) | Compartment c VSS (g/L) | Compartment d TSS (g/L) | Compartment d VSS (g/L) | Effluent TSS (g/L) | Effluent VSS (g/L) |
|---|---|---|---|---|---|---|---|---|---|---|
| 1st | 0.98 | 0.79 | 0.63 | 0.43 | 0.57 | 0.33 | 1.16 | 0.99 | 0.75 | 0.63 |
| 2nd | 0.61 | 047 | 0.64 | 0.52 | 0.60 | 0.54 | 0.63 | 0.54 | 0.58 | 0.49 |
| 3rd | 0.50 | 0.31 | 0.44 | 0.26 | 0.55 | 0.35 | 0.50 | 0.31 | 0.48 | 0.32 |
| 4th | 0.50 | 0.35 | 0.43 | 0.31 | 0.46 | 0.31 | 0.50 | 0.32 | 0.48 | 0.32 |
| 5th | 0.30 | 0.20 | 0.32 | 0.31 | 0.33 | 0.23 | 0.34 | 0.21 | 0.45 | 0.28 |
| 6th | 0.12 | 0.24 | 0.20 | 0.13 | 0.24 | 0.13 | 0.20 | 0.15 | 0.25 | 0.16 |

*3.2. Second Experimental Cycle*

The average results and their standard deviation for every phase of the second experimental cycle are presented in Table 6.

**Table 6.** Average daily results of the second experimental cycle phases.

| Second Experimental Cycle | Experimental Phases | | | |
|---|---|---|---|---|
| Characteristics | 1st | 2nd | 3rd | 4th |
| Biogas Production (L/d) | $58.1 \pm 3.1$ | $68.0 \pm 4.1$ | $71.6 \pm 8.6$ | $66.8 \pm 2.3$ |
| Biomethane Production (L/d) | $42.9 \pm 7.4$ | $44.6 \pm 6.4$ | $42.3 \pm 4.5$ | $41.1 \pm 3.5$ |
| $CH_4$ Biogas content (%) | $71.7 \pm 11.2$ | $65.07 \pm 7.5$ | $60.01 \pm 5.3$ | $61.6 \pm 4.7$ |
| sCOD Feedstock (g/L) | $3.89 \pm 0.10$ | $2.92 \pm 0.30$ | $2.05 \pm 0.12$ | $1.03 \pm 0.15$ |
| sCOD Effluent (g/L) | $0.82 \pm 0.04$ | $0.48 \pm 0.17$ | $0.24 \pm 0.10$ | $0.27 \pm 0.06$ |
| COD removal (%) | 78.9 | 83.5 | 88.3 | 73.7 |

In Figure 11, the daily production of biogas and biomethane in the reactor per experimental phase shows increase in biogas production after the first experimental phase and then a significant reduction in the beginning of the third experimental phase. In the middle of the third experimental phase, the reactor increased again its biogas generation, and during the 56th day of operation reached the maximum production recorded (81.3 L biogas). During the last experimental phase (HRT 1 d) the reactor showed greater stability both in terms of biogas and biomethane production. The same pattern is shown in Figure 12, where the daily methane percentage in the generated biogas is depicted.

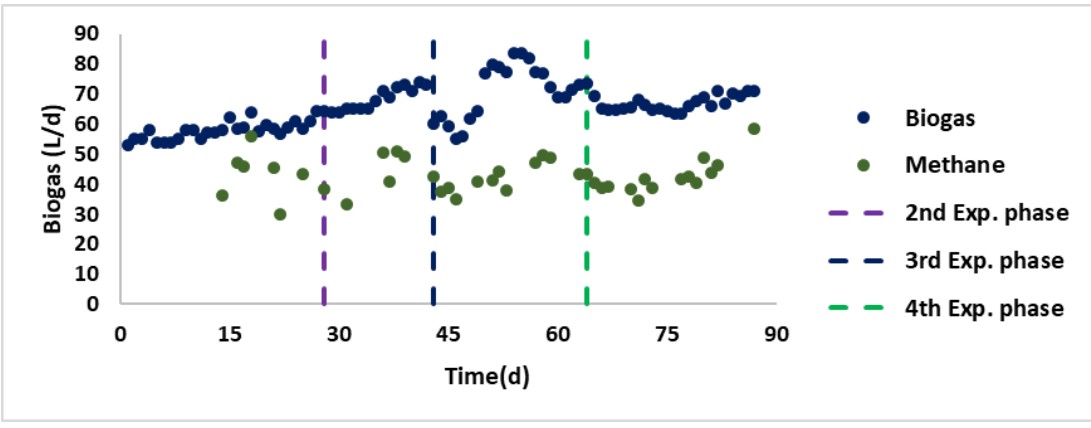

**Figure 11.** Daily biogas and biomethane production.

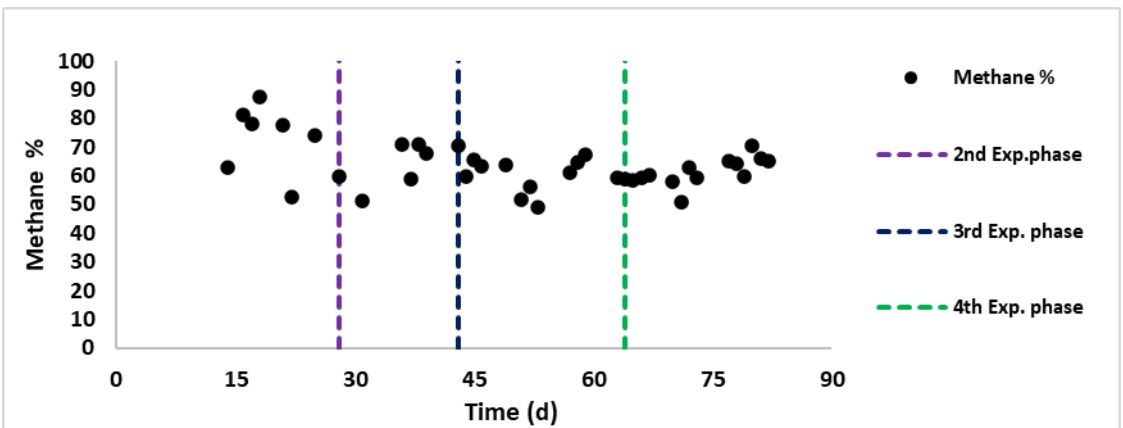

**Figure 12.** Daily methane concentration of biogas generated from the PABR.

Regarding the COD consumption in this experimental cycle, again a significant reduction was presented in the reactor's effluent. Although reduction levels reached 88.3% (2d HRT), the effluent still does not reach the criteria for disposal. The lower effluent COD concentration achieved is that of 240 mg/L in the third experimental phase and the highest value in COD effluent was observed during the first experimental phase 820 mg/L. The daily sCOD values of each compartment are presented in Figure 13, while the daily sCOD values of the feedstock and the effluent are presented in Figure 14. The sCOD concentration of the effluent in the last experimental phase shows that a significant reduction is obtained (73.7%). However, as in the previous cycle, the effluent is not directly disposable.

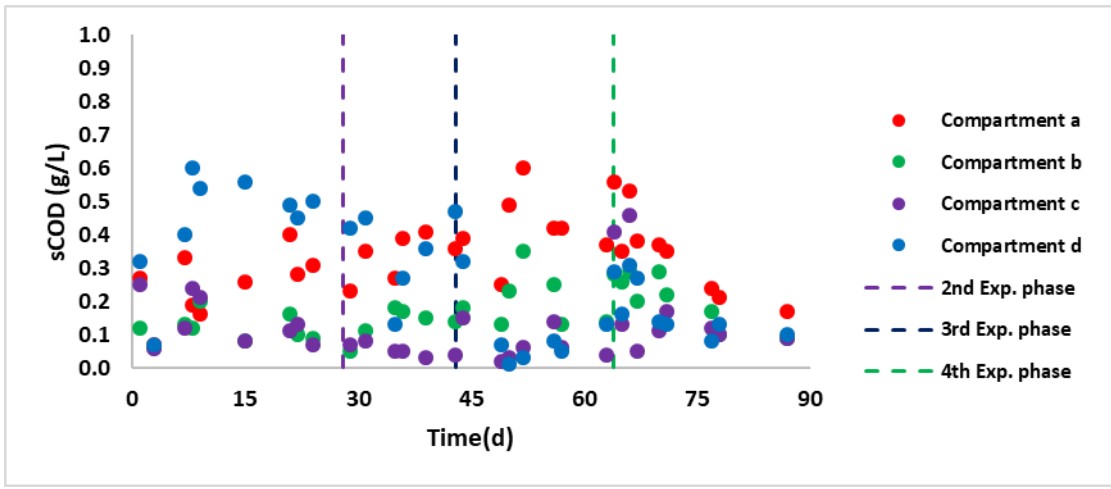

**Figure 13.** Daily sCOD values of PABR compartments.

The pH of the process observed in Figure 15 was fairly constant during all experimental phases of the cycle for all compartments, without any significant fluctuation. Also, by adding NaHCO$_3$ as alkalinity buffer in the feedstock, the alkalinity of the process remained at the desired levels of 2000–2500 mgCaCO$_3$/L for all experimental phases, as shown in Figure 16. The average daily pH and total alkalinity values are presented in Table 7.

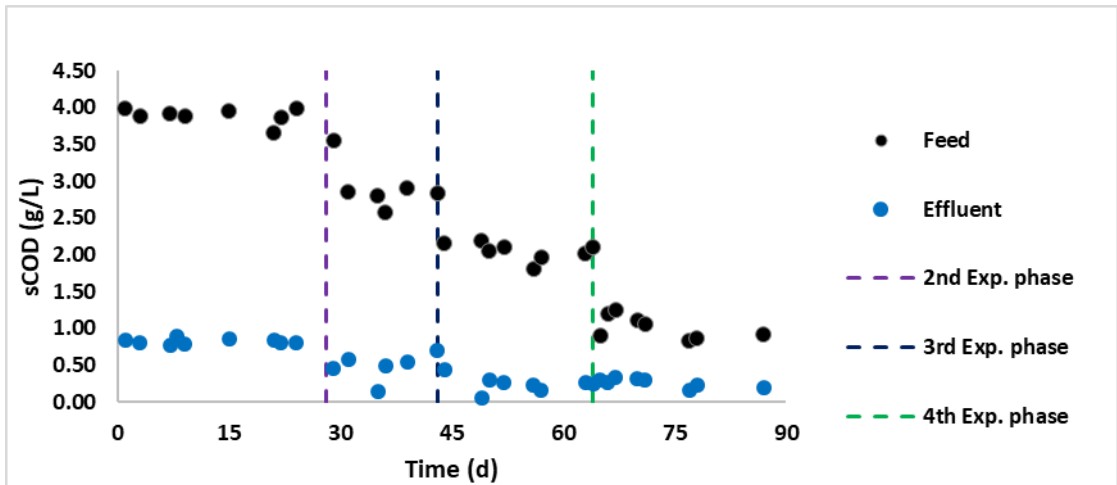

**Figure 14.** Daily sCOD values of the reactor's feedstock and effluent.

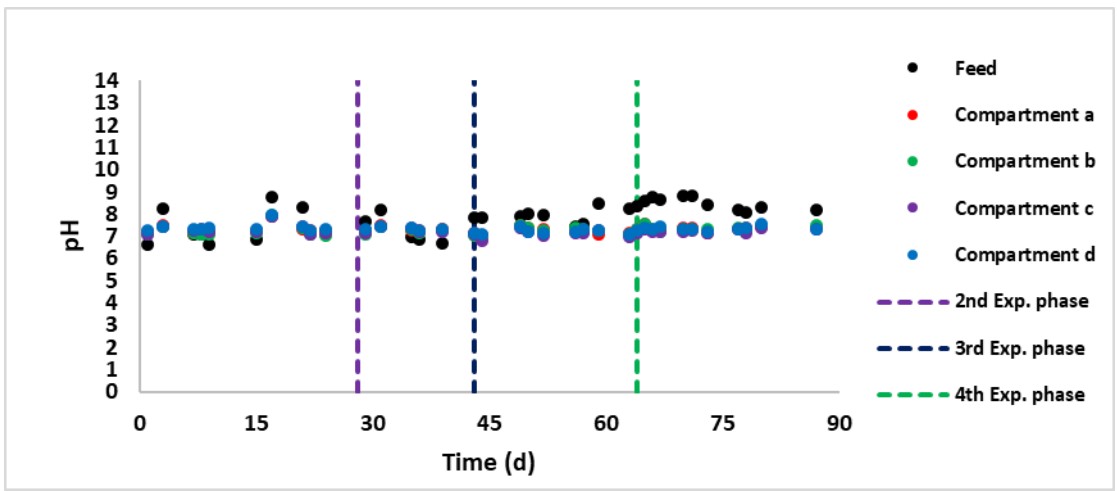

**Figure 15.** Daily pH values of the feedstock and the reactor compartments.

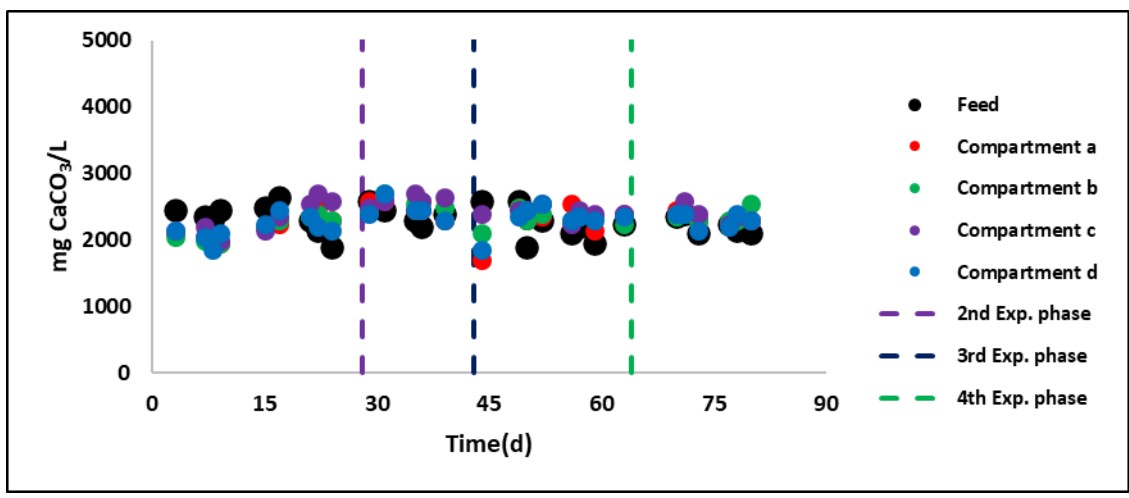

**Figure 16.** Daily total alkalinity value of the feedstock and the reactor's compartments.

**Table 7.** Average daily values of pH and total alkalinity of the reactor's feedstock, compartments and effluent.

| | Feedstock | | Compartment a | | Compartment b | | Compartment c | | Compartment d | | Effluent | |
|---|---|---|---|---|---|---|---|---|---|---|---|---|
| Exp. Phase | pH | Alkalinity mgCaCO₃/L | pH | Alkalinity mgCaCO₃/L | pH | Alkalinity mgCaCO₃/L | pH | Alkalinity mgCaCO₃/L | pH | Alkalinity mgCaCO₃/L | pH | Alkalinity mgCaCO₃/L |
| 1st | 7.41 | 2339 | 7.30 | 2188 | 7.40 | 2172 | 7.31 | 2294 | 7.25 | 2172 | 7.41 | 2100 |
| 2nd | 7.37 | 2390 | 7.24 | 2480 | 7.31 | 2560 | 7.29 | 2610 | 7.22 | 2460 | 7.31 | 2410 |
| 3rd | 7.98 | 2238 | 7.14 | 4806 | 7.22 | 2319 | 7.25 | 2419 | 7.28 | 2312 | 7.47 | 2287 |
| 4th | 8.51 | 2225 | 7.24 | 2333 | 7.37 | 2367 | 7.36 | 2383 | 7.37 | 2308 | 7.60 | 2375 |

The average amount of TSS and VSS per compartment at every experimental phase is presented in Table 8, while their daily concentrations are presented in Figures 17 and 18, respectively.

**Table 8.** Average daily values of reactor's compartment and effluent.

| Exp. Phase | Compartment a | | Compartment b | | Compartment c | | Compartment d | | Effluent | |
|---|---|---|---|---|---|---|---|---|---|---|
| | TSS (g/L) | VSS (g/L) | TSS (g/L) | VSS (g/L) | TSS (g/L) | VSS (g/L) | TSS (g/L) | VSS (g/L) | TSS (g/L) | VSS (g/L) |
| 1st | 0.38 | 0.26 | 0.24 | 0.17 | 0.24 | 0.17 | 0.43 | 0.33 | 0.48 | 0.34 |
| 2nd | 0.39 | 0.34 | 0.30 | 0.21 | 0.22 | 0.21 | 0.49 | 0.40 | 0.52 | 0.38 |
| 3rd | 0.66 | 0.43 | 0.31 | 0.21 | 0.26 | 0.17 | 0.40 | 0.27 | 0.26 | 0.19 |
| 4th | 0.32 | 0.23 | 0.19 | 0.11 | 0.16 | 0.11 | 0.30 | 0.23 | 0.18 | 0.13 |

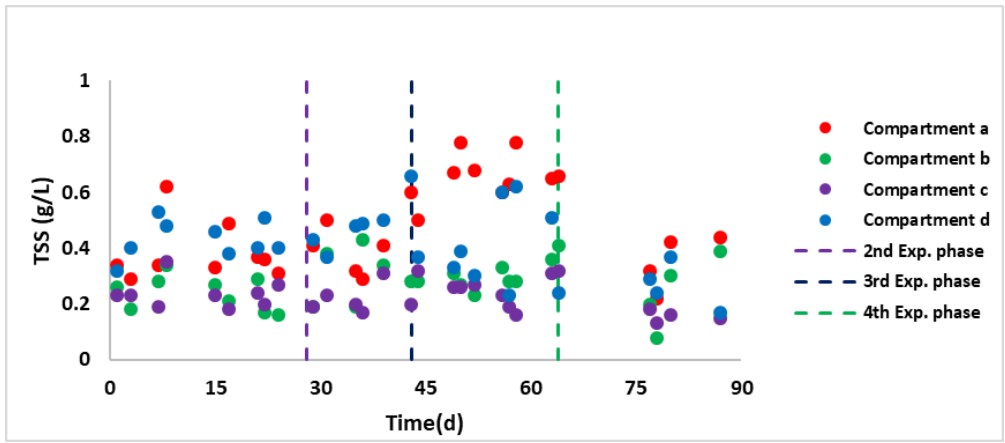

**Figure 17.** Daily TSS concentration in the reactor's compartments.

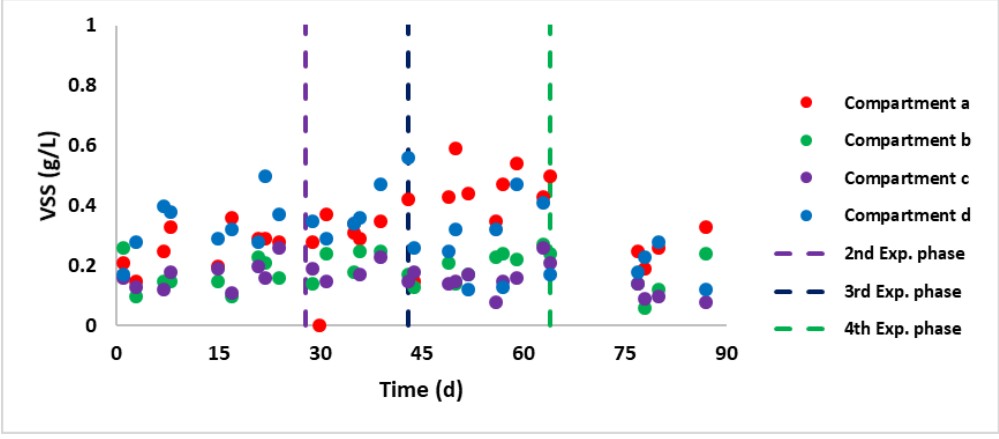

**Figure 18.** Daily VSS concentration in the reactor's compartments.

Comparing the two operational modes, it is concluded that in both switching and ABR mode, the process shows significant efficiency, especially during the last phase of each cycle, where the feedstock resembles the most the composition of municipal wastewater. In terms of possible energy recovery, during the sixth phase of the first cycle biogas generation reached 44.3 L/d with almost 60% methane content, while during the fourth phase of the second cycle with similar conditions imposed, biogas generation reached 66.8 L/d and methane content 61.5%. On the other hand, in terms of waste treatment, during the last phases of the first and second cycles, the COD removal rate reached 64.9% and 73.7%, respectively. In both aspects the ABR operational mode presents better results.

## 4. Conclusions

In this paper, we evaluated the efficiency of a PABR for the treatment of synthetic municipal wastewater operating under different conditions and assessed biogas and biomethane productivity along with COD removal. During both experimental cycles, the reactor showed high potential both in terms of biomethane generation and of organic matter reduction. The most significant results were obtained during the last phase of each experimental cycle, where the feedstock resembled the most the municipal wastewater characteristics.

During the PABR operation (first experimental cycle) the process was efficient in terms of energy recovery, but the organic content of the effluent in the last experimental phase was higher than the environmental limit. During the ABR operation (second experimental cycle), biogas and biomethane production reached 66.8 L/d and 41.1 L/d, respectively. However, similarly to the first experimental cycle, the COD concentration of the reactor's effluent is over the environmental limit.

As a conclusion, the PABR is a high-rate anaerobic digestion system capable of operating in low organic loadings and low HRTs. This capability gives the PABR the potential of reducing the energy consumption of MWW management by operating as a pretreatment step before the aeration tank in MWW treatment plants.

**Author Contributions:** Methodology, A.Z., C.K., A.M., K.P. and G.L.; investigation, G.L. and K.P.; writing—original draft preparation, A.Z., K.P. and G.L.; writing—review and editing, G.L. and K.P.; supervision, K.P. and G.L. All authors have read and agreed to the published version of the manuscript.

**Funding:** Hellenic Foundation for Research and Innovation (H.F.R.I.) under the "First Call for H.F.R.I. Research Projects to support Faculty members and Researchers and the procurement of high-cost research equipment grant" (project 2797).

**Institutional Review Board Statement:** Not applicable.

**Informed Consent Statement:** Not applicable.

**Data Availability Statement:** Not applicable.

**Conflicts of Interest:** The authors declare no conflict of interest. The funders had no role in the design of the study, the collection, analyses, or interpretation of data, the writing of the manuscript, or the decision to publish the results.

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
