# Peer review of "Anaerobic Digestion of Synthetic Municipal Wastewater (MWW) in a Periodic Anaerobic Baffled Reactor (PABR): Assessment of COD Removal and Biogas Production"

_applsci, doi:10.3390/app122413037_

Round 1

Reviewer 1 Report

The manuscript entitled “Direct Anaerobic Digestion of Municipal Wastewater in a Periodic Anaerobic Baffled Reactor (PABR)” is informative. I have some comments for the manuscript improvement. My comments are as follows,

1.      The abstract part is too lengthy. I would suggest the authors to concise it.

2.      In abstract part, Line number -11, the authors mentioned the term “innovative”. In which way the authors claiming that the anaerobic baffled reactor is novel? There are several research indicating the functions of ABR. Kindly clarify it.

3.      The sentences starts with “The aim of this work is to examine…” in line no-59 was repeating in abstract and introduction part.

Author Response

We would like to kindly thank you for your time and effort reviewing our paper entitled ‘’Anaerobic Digestion of Synthetic Municipal Wastewater (MWW) in a Periodic Anaerobic Baffled Reactor (PABR): Assessment of COD Removal and Biogas Production’’.

Please accept our revised version of the paper, in which we have tried to address all the comments provided by you, aiming at significantly improving the quality and contribution of our paper.

Reviewer 1

Q1.      The abstract part is too lengthy. I would suggest the authors to concise it.

A1. The abstract has been revised

Q2.      In abstract part, Line number -11, the authors mentioned the term “innovative”. In which way the authors claiming that the anaerobic baffled reactor is novel? There are several research indicating the functions of ABR. Kindly clarify it.

A2. The innovation of this work is based on using a PABR reactor not a regular ABR. By using the PABR it is possible to switch flow patterns between that of an UASBR and that of an ABR with four consecutives compartments. Also the approach of direct anaerobic digestion of municipal wastewater is different from the benchmark approach as it reduces the energy requirements and the waste generated from the overall process.

Q3.      The sentences starts with “The aim of this work is to examine…” in line no-59 was repeating in abstract and introduction part.

A3. The repeated sentence has been revised in the whole text

Reviewer 2 Report

File attached

Author Response

We would like to kindly thank you for your time and effort reviewing our paper entitled ‘’Anaerobic Digestion of Synthetic Municipal Wastewater (MWW) in a Periodic Anaerobic Baffled Reactor (PABR): Assessment of COD Removal and Biogas Production’’.

Please accept our revised version of the paper, in which we have tried to address all the comments provided by you, aiming at significantly improving the quality and contribution of our paper.

Reviewer 2

Q1. Title may be modified according to results as results are showing biogas production too. Also it can be depicted from conclusion that we evaluated the efficiency of a PABR for the treatment of a synthetic municipal wastewater operating under different conditions and assessed biogas and bio-methane productivity along with COD removal.

A1. The title has been revised

Q2. Abstract need improvement. It must be in logical sequence like: Background, problem/gap, objectives, methods, Results, conclusions, future recommendations if any.

A2. Abstract has been revised

Q3. In section 2.2: Composition of medium and other relevant information should be shifted to the relevant section of Methods. This section is only for analysis methods.

A3. A new section has been created and the paragraph has been shifted

Q4. Title from diagrams should be removed as it is already mentioned in caption.

 A4. Titles from the diagrams have been removed

Q5. Acknowledgments section must be included.

A5. Acknowledgment has been added
